# Exploration-Based Planning for Multiple-Target Search with Real-Drone Results

**DOI:** 10.3390/s24092868

**Published:** 2024-04-30

**Authors:** Bilal Yousuf, Zsófia Lendek, Lucian Buşoniu

**Affiliations:** Department of Automation, Technical University of Cluj-Napoca, Memorandumului 28, 400114 Cluj-Napoca, Romania; zsofia.lendek@aut.utcluj.ro (Z.L.); lucian.busoniu@aut.utcluj.ro (L.B.)

**Keywords:** multi-target search, probability hypothesis density filter, Parrot Mambo minidrone, exploration-based search

## Abstract

Consider a drone that aims to find an unknown number of static targets at unknown positions as quickly as possible. A multi-target particle filter uses imperfect measurements of the target positions to update an intensity function that represents the expected number of targets. We propose a novel receding-horizon planner that selects the next position of the drone by maximizing an objective that combines exploration and target refinement. Confidently localized targets are saved and removed from consideration along with their future measurements. A controller with an obstacle-avoidance component is used to reach the desired waypoints. We demonstrate the performance of our approach through a series of simulations as well as via a real-robot experiment in which a Parrot Mambo drone searches from a constant altitude for targets located on the floor. Target measurements are obtained on-board the drone using segmentation in the camera image, while planning is done off-board. The sensor model is adapted to the application. Both in the simulations and in the experiments, the novel framework works better than the lawnmower and active-search baselines.

## 1. Introduction

We consider a drone exploring a 3D environment to find an unknown number of static targets at unknown locations as quickly as possible. Examples of such target-search problems include search and rescue [1], monitoring activities [2], exploration of unknown places for hazards [3], etc. The specific problem motivating our approach is the search for underwater litter: see https://seaclear-project.eu (accessed on 1 March 2024). The main goal is to reduce the trajectory length (number of steps) since, in practice, real-world robot motion and sensor samples are often the most costly resources. The drone is equipped with an imperfect sensor that has a limited field of view (FOV), may miss targets, and takes noisy measurements of the detected targets. The FOV is the extent of the scene visible by the sensor from a given position, and is modeled using a position-dependent probability of detection. Based on the measurements, a sequential Monte Carlo–probability hypothesis density (SMC-PHD) multi-target filter [4] is run in the framework of random finite sets [5,6]. The filter uses weighted particles to represent an intensity function, which is a generalization of the probability density that integrates, not to a probability mass, but to the expected number of targets [5].

Many ways have been proposed to search for and identify a known or unknown number of dynamic or static targets from measurements taken by mobile agents in robotics [7,8,9,10,11,12,13,14,15], control [16,17,18,19], reinforcement learning [20,21,22,23,24,25,26,27,28,29], multi-target filtering [5,12,13,14,18,23,30,31,32,33,34], etc. Among all these fields, we focus here on multi-target filtering with an intensity function representation because this framework best accommodates our setting. Approaches in other fields use different models of uncertainty (e.g., uncertainty on the robot’s pose and location on the map, versus our case of an unknown number of targets at unknown locations that are unreliably and noisily detected) and different representations, like occupancy maps, which are less suited to our setting than intensity functions. For instance, targets clustered in a single cell of an occupancy map will not be identified properly, and if the number of targets is not very large, an occupancy grid may be inefficient. From the point of view of path planning and exploration, our main contribution is to formulate a search potential to be maximized and a corresponding strategy such that all targets are found with sufficient accuracy and in minimum time given the uncertainty concerning the number of targets and detecting targets and available noisy measurements.

In multi-target filtering, most methods use only variants of mutual information (MI) to evaluate the potential waypoints of the agents, e.g., [5,18,33]. MI measures the amount of information between agent trajectories and the event of not seeing any targets, so maximizing MI decreases the chances of not seeing previously observed targets (in other words, it increases the chances of seeing them). A key shortcoming of such methods is that exploration of the environment to find new targets is—to our best knowledge—not explicitly considered. Instead, some methods include a search component for an unknown-target intensity function [35,36,37], which can be seen as an indirect form of exploration. Methods from other fields do explore or even aim to search and map the whole environment [10,11,12,13,14,15,21,25,27,38,39], but, as explained above, they are unsuitable in our setting.

We therefore propose here the first method to search for an unknown number of targets based on intensity functions that *explicitly explores for new targets*. In this method, a new drone path planner selects future positions by maximizing a finite-horizon objective function that balances two components: exploration and target refinement, somewhat similar to the basic behaviors and the corresponding potential fields in, e.g., [12,40,41]. The exploration component drives the drone toward unseen regions of space. It is implemented using an exploration bonus that is initialized to 1 over the entire environment and then is decreased in the observed regions. Target refinement aims to better locate targets for which measurements were previously received and is computed in one of two ways. The first option is standard MI, like in [18,30,31,32,33,42]. Moreover, since evaluating MI requires expensive computation, we introduce a computationally cheaper option than MI, in which the probabilities of detection at estimated target locations are summed up.

As a result of maximizing the objective function, the planner returns a sequence of waypoints, the first of which is implemented using a controller that also performs obstacle avoidance. The procedure is repeated for the receding horizon. Estimated target locations are computed as the centers of K-means clusters of particles [43,44,45]. Narrow enough clusters that contain a large enough intensity mass are declared as found targets To prevent re-identifying found targets, we remove their corresponding particles and future measurements that are likely to originate from them.

The proposed method is extensively validated in a series of simulations and experiments that are concerned with finding an unknown number of targets. We first study the influence of the planning horizon and find that horizon 1 suffices in practice. We then pitch our method against three baselines: a predefined lawnmower pattern that uniformly covers the environment [46], compared to which we aim to reduce the number of robot steps required to find all the targets, and two active-search methods representative of the multi-target filtering field: an MI-only method without any exploration, adapted from [33], and a method with an unknown-target search component, adapted from [35]. For both uniformly distributed and clustered targets, our method finds all targets faster than these baselines. Using the full exploration-based objective, we then compare MI to our center-probability technique and find that computational cost is reduced without introducing any statistically significant difference in target detection performance. A key contribution is our real-life experiment, which involves a Parrot Mambo minidrone that searches from a constant altitude for targets on the floor. Here, our new method again works better than the lawnmower and the MI-only methods, confirming the simulation results.

Summarizing, the key contributions of our paper include:A method to search for an unknown number of static targets at unknown positions that uses an intensity-function multi-target filter to handle highly uncertain sensors and is the first to combine such a filter with an explicit exploration objective in the planner;Detailed simulation results in which we compare our method to three baselines, including a lawnmower and two active search methods;A real experiment involving a Parrot Mambo that searches indoors for targets located on the floor and via which we compare our method to lawnmower and active-search methods.

This paper is a heavily extended and revised version of our preliminary conference version [47], with the following additional elements: (i) validation via real-life experiments using a Parrot Mambo minidrone (manufactured by Parrot SA, Paris, France) (ii) extension of the planner to a receding horizon, whereas it was myopic (single-step) before, (iii) generalization from 2D to 3D, and (iv) the addition of an obstacle avoidance strategy to the control of the drone.

Next, Section 2 formulates the problem, followed by background information about PHD filtering in Section 3. Section 4 describes the proposed method. Section 5 presents simulation results, followed by the hardware setup and experimental results in Section 6. Section 7 concludes the paper.

## 2. Problem Formulation

The problem we are addressing concerns a drone that navigates a 3D space (environment) *E* containing an initially unknown number of stationary targets, as depicted in Figure 1 (left). The primary goal is to identify the positions of all the targets as accurately as possible using a trajectory that is as short as possible. The main challenge is that target sensing is affected by two sources of uncertainty: missed targets due to a probabilistic field of view and measurement noise for those targets that are detected: see the sensor model below. A planner will be formulated in Section 4.1 that aims to both find new targets and to reduce uncertainty about seen targets.

The drone’s dynamics are defined by:(1)$$q_{t+1}=f\left(q_{t},u_{t}\right)$$
where *t* represents the discrete time step. The input $u_{t}$ can be, for example, a state feedback control law:(2)$$u_{t}=h\left(q_{t}\right),$$
but various other controllers can be employed. It is assumed that the drone’s state $q_{t}\;\in \;E$ is known accurately enough to avoid the need for an explicit treatment of state uncertainty, and that for any ϵ>0, the controller can maneuver the drone to within an ϵ-neighborhood of any location in *E* in a finite number of steps.

At the given discrete time step *k*, a set $X_{k}$ contains $N_{k}$ stationary targets at positions $x_{ik}\;\in \;E,\;\;\;i=1,2,\ldots ,\;N_{k}$. Note that the drone’s dynamics have a time scale (time step *t*) that different from that of the measurement and planning (time step *k*). Both the cardinality $N_{k}$ and the locations of the targets are initially unknown. Although we assume that the real set of targets is static, new targets are seen over time, so a time-dependent notation is employed.

The sensor model describes how measurements are taken. Sensor uncertainty manifests in two ways. Firstly, at each step, the sensor does not see all the targets but may miss some depending on a probabilistic FOV. Secondly, the measurements of seen targets are affected by noise.

In general, the probability with which the drone at position *q* detects a target at position *x* is denoted by π(x,q). For example, in our simulations, we will consider an omnidirectional ranging sensor [2,3,19,33,48] for which the probability is defined as:(3)$$\pi \left(x,q\right)=Ge^{-∥\zeta ∥/2}$$
where scalar G≤1, and:$$\zeta =\left(\frac{X_{x}-X_{q}}{F_{X}},\frac{Y_{x}-Y_{q}}{F_{Y}},\frac{Z_{x}-Z_{q}}{F_{Z}}\right)$$
is a normalized distance between the target and the sensor (drone). In this expression, $\left(X_{x},Y_{x},Z_{x}\right)$ is the 3D position of the target, $\left(X_{q},Y_{q},Z_{q}\right)$ are the 3D coordinates of the drone position *q*, and $\left(F_{X},F_{Y},F_{Z}\right)$ are normalization constants that may be interpreted as the size of the (probabilistic) FOV. For example, when these constants are all equal, π is radially symmetric around the drone position, as illustrated in Figure 1 (right). The planner works for other forms of π, and, in fact, for the real drone, the FOV will be different: see Equation (Equation 19).

The binary event $b_{ik}$ of detecting a target $x_{ik}$ then naturally follows a Bernoulli distribution given by the probability of detection at *k*: $b_{ik}∼B\left(\pi \left(x_{ik},q_{k}\right)\right)$. Given these Bernoulli variables and the actual target positions $\left(X_{x_{ik}},Y_{x_{ik}},Z_{x_{ik}}\right)$ in the space *E*, the set of measurements $Z_{k}$ is:(4)$$Z_{k}=\underset{i\;\in \left\{1,\ldots ,N_{k}\right\}\;s.t.\;b_{ik}=1}{⋃}\left[g_{ik}\left(x_{ik}\right)+\varrho_{ik}\right]$$
where $g_{ik}\left(x_{ik}\right)$ is defined as:$$\begin{array}{cc} \; & g_{ik}\left(x_{ik}\right)={\left[d_{ik},\theta_{ik},\varpi_{ik}\right]}^{T}\\ \; & d_{ik}=\sqrt{{\left(X_{x_{ik}}-X_{q_{k}}\right)}^{2}+{\left(Y_{x_{ik}}-Y_{q_{k}}\right)}^{2}+{\left(Z_{x_{ik}}-Z_{q_{k}}\right)}^{2}},\\ \; & \theta_{ik}=\arctan\frac{Y_{x_{ik}}-Y_{q_{k}}}{X_{x_{ik}}-X_{q_{k}}},\;\;\varpi_{ik}=\arcsin\frac{Z_{x_{ik}}-Z_{q_{k}}}{d_{ik}} \end{array}$$
So, for each target that is detected, the measurement consists of a range $d_{ik}$, bearing angle $\theta_{ik}$, and elevation angle $\varpi_{ik}$ with respect to the drone. This measurement is affected by Gaussian noise $\varrho_{ik}∼N\left(.,0,R\right)$ with mean $0={\left[0,0,0\right]}^{⊤}$ and diagonal covariance $R=\mathrm{diag}(\sigma^{2},\sigma^{2},\sigma^{2})$. Thus, the target measurement density is:(5)$$p\left(z_{k}|x\right)=N\left(z_{k},g_{k}\left(x\right),R\right),$$
i.e., it is a Gaussian density function with covariance *R* centered on g(x). This density will be used to estimate the target locations.

## 3. Background on PHD Filtering

The probability hypothesis density (PHD) D:E→[0,∞), or intensity function, is similar to a probability density function, with the key difference being that its integral $\int_{S}D\left(x\right)dx$ over some subset S⊆E is the expected number of targets in *S* instead of the probability mass of *S*, as it would be for a density.

The PHD filter [4] performs Bayesian updates of the intensity function based on the target measurements and is summarized as:(6)$$\begin{array}{cc} \; & D_{k|k-1}=\Phi_{k|k-1}\left(D_{k-1|k-1}\right)\\ \; & D_{k|k}=\Psi_{k}\left(D_{k|k-1},Z_{k}\right) \end{array}$$
Here, $D_{k|k-1}$ is the prior intensity function predicted based on intensity function $D_{k-1|k-1}$ at time step k−1, and $D_{k|k}$ denotes the posterior generated after processing the measurements. The multi-target prior $D_{k|k-1}$ at step *k* is defined by:(7)$$D_{k|k-1}\left(x_{k}\right)=\Phi_{k|k-1}\left(D_{k-1|k-1}\right)\left(x_{k}\right)=\Upsilon +\int_{E}p_{s}\left(\xi \right)\delta_{\xi }\left(x_{k}\right)D_{k-1|k-1}\left(\xi \right)d\xi$$
where $p_{s}\left(\xi \right)$ is the probability that an old target at position ξ still exists, and the transition density of a target at ξ is defined as the Dirac delta $\delta_{\xi }\left(x\right)$ centered on ξ. This is because in our case, targets are stationary. Finally, Υ denotes the intensity function of a new target appearing and is chosen here as a constant. The posterior intensity function $D_{k|k}$ at step *k* using the measurements $Z_{k}$ is computed as:(8)$$\begin{array}{cc} \; & D_{k|k}\left(x_{k}\right)=\Psi_{k}\left(D_{k|k-1},Z_{k}\right)\left(x_{k}\right)=\left[1-\pi \left(x_{k},q_{k}\right)+\sum_{z\;\in \;Z_{k}}\frac{\psi_{kz}\left(x_{k}\right)}{\langle \psi_{kz},D_{k|k-1}\rangle \left(x_{k}\right)}\right]\cdot D_{k|k-1}\left(x_{k}\right) \end{array}$$
where $\psi_{kz}\left(x_{k}\right)=\pi \left(x_{k},q_{k}\right)p\left(z_{k}|x_{k}\right)$ denotes the overall probability of detecting a target at $x_{k}$, with *p* defined in (Equation 5), and $\langle \psi_{kz},D_{k|k-1}\rangle =\int_{E}\psi_{kz}\left(x_{k}\right)D_{k|k-1}\left(x_{k}\right)dx_{k}$. In practice, we employ the SMC-PHD filter [4], which uses at each step *k* a set of weighted particles $\left(x_{k}^{i},\omega_{k|k}^{i}\right)$ to represent $D_{k|k}$, with the property that $\int_{S}D_{k|k}\left(x\right)dx\approx \sum_{x_{k}^{i}\in S}\omega_{k|k}^{i}$ for any S⊆E. For more details about the particle-based implementation, see Appendix A.

An example of an intensity function in 2D is given in Figure 2, where the three peaks correspond to possible target locations, and the circles illustrate the weighted particles. Note that in reality, there is no constraint that particle weights are on the PHD surface; this situation is shown here to give a more intuitive representation. The red patch in Figure 2 is the intensity function *D* defined over the corresponding rectangle *S* lying in the (X,Y) plane. The integral of *D* over this red region gives the expected number of targets in *S*.

The PHD filter will be used by the planner in the following section to estimate target locations from measurements.

## 4. Exploration-Based Search

This section presents the main contribution of the paper: the novel target search algorithm, which is organized in three components. First, the proposed drone path planner is described in Section 4.1. Second, in Section 4.2, we present a method to mark well-defined targets as found and disregard measurements likely to come from these targets in the future. Finally, an obstacle avoidance mechanism is given in Section 4.3.

### 4.1. Planner

Consider first the problem of designing a 3D path to follow so as to find the targets. A classical solution to this problem would be a 3D variant of a lawnmower trajectory that fills the space in a uniform manner. We evaluate the lawnmower as a baseline in our experiments, but a solution that finds the targets more quickly is desired. We next propose a receding-horizon path planner that generates such a solution as a sequence of waypoints for the drone to track and, in addition to exploring the space with the goal of finding all targets, focuses on refining potential targets that were already (but poorly) measured.

At each step *k*, a new waypoint is generated by the planner. To formalize the planner, first define the integer horizon τ>0 and a potential sequence of next positions of the robot $q_{k}=\left(q_{k+1},q_{k+2},\ldots ,q_{k+\tau }\right)$. In this sequence, each potential next position is defined relative to the previous one:(9)$$q_{k+j+1}=q_{k+j}+\delta q_{j},\;\mathrm{for}\;j=0,\ldots ,\tau -1$$
where the set of possible position changes δq is discrete and should be sufficiently rich to find the targets. The sequence of next positions is determined by solving the following optimization problem:(10)$$q_{k}^{*}\in {\mathrm{argmax}}_{q_{k}}\left\{\alpha \cdot E\left(q_{k}\right)+T\left(q_{k}\right)\right\}$$
The objective function has two components: exploration $E(q_{k})$ and target refinement $T(q_{k})$, with α being a tunable parameter that controls the tradeoff between the two components; a larger α emphasizes exploration more. The first of the positions found by (Equation 10) will be the next waypoint of the drone, and then, the optimization procedure is repeated, leading overall to a receding-horizon scheme similar to model-predictive control [49].

The *exploration component* $E(q_{k})$ of (Equation 10) is novel and drives the robot to look at unseen regions of the environment. To achieve this, first define an exploration bonus function ι, which is initialized to 1 for the entire environment and decreases at each step *k* and each position *x* by an amount related to $\pi (x,q_{k})$. The meaning is that each position *x* has been explored to an amount related to the probability of detection at that position. To implement the exploration bonus, we represent ι on a 3D grid of points $x_{ijl}$, which is initialized with:$$\iota_{0}\left(x_{ijl}\right)=1,\forall i,j,l$$
and updated with:(11)$$\iota_{k}\left(x_{ijl}\right)=\iota_{k-1}\left(x_{ijl}\right)\cdot \left(1-\pi \left(x_{ijl},q_{k}\right)\right),\forall i,j,l,\forall k\geq 1$$
Then, the exploration component is defined as:(12)$$E\left(q_{k}\right)=\sum_{j=1}^{\tau }\sum_{i=1}^{c_{k}}\iota_{k}\left(q_{k}\right)$$
where $\iota_{k}$ at positions $q_{k}$ that are not on the grid are computed by trilinear interpolation.

The *target refinement* component $T(q_{k})$ in (Equation 10) focuses on refining the locations of targets about which measurements were already received by driving the robot to areas where the intensity function is larger. The refinement component can be computed in two ways.

The first option is the MI between the targets and the empty measurement set along the horizon, which we use and compute as in [33]. Since this MI is maximized, the event of receiving empty measurements is expected to become low-probability. Note that since probabilities of detection depend on the position sequence $q_{k}$, the MI also depends on these positions. A shortcoming of MI is that it is computationally expensive due to the need to compute the entropy of the target intensity function and the conditional entropy between the next position of the agent and a binary measurement event.

We therefore propose a second, computationally more efficient alternative. At each step *k*, we extract potential targets as clusters of particles generated with K-means [4]. The target refinement component then consists of the sum of the observation probabilities of all cluster centers that have accumulated along the horizon:(13)$$T\left(q_{k}\right)=\sum_{j=1}^{\tau }\sum_{i=1}^{c_{k}}\pi \left({\hat{x}}_{i,k},q_{k+j}\right)$$
where the probability of detection π was defined in (Equation 3), $c_{k}$ denotes the number of clusters at *k*, and ${\hat{x}}_{i,k}$ is the center of the *i*th cluster $C_{i,k}$. This center has the meaning of an estimated target position. This option will be called “center probabilities”, for short. The intuition is that the probability of observing estimated target locations is maximized.

Note that in classical target searching [30,31,32,33,42], only MI target refinement is used, which means that the robot only focuses on targets that it already saw, without exploring for new targets. Conversely, when no targets have been seen yet (or when all seen targets have been marked as found, see below for details), planner (Equation 10) will compute the robot’s trajectory solely based on the exploration component, so a space-filling lawnmower-like trajectory is obtained: see experiment E6 in Section 5 for an example. In most cases, both objectives are important, and the proposed planner strikes a balance between them as controlled by the tuning parameter α.

### 4.2. Marking and Removal of Found Targets

Even when certain targets are well-determined (i.e., they have clear peaks in the intensity function), the target refinement component will still focus on them, which is not beneficial since the time would be better spent refining poorly seen targets or looking for new targets. To achieve this, the algorithm removes such well-determined targets, as explained next. After a measurement is processed, we extract potential targets as clusters of particles with K-means [4]. Then, each cluster that is narrow enough and is associated with a large enough concentration of mass in the intensity function is taken to correspond to a well-determined, or *found*, target. Two conditions are checked: that the cluster radius $r_{i,k}$ is below a threshold $T_{r}$ and that the sum of the weights $\omega_{j}$ of the particles in the cluster is above a mass threshold $T_{m}$. The center ${\hat{x}}_{i,k}$ of such a cluster is added to a set $\hat{X}$ of found targets, and the particles belonging to that cluster are deleted.

To prevent the formation of another intensity peak at the locations of old, already-found targets, measurements likely to be associated with these targets are also removed from future measurement sets. Of course, the algorithm has no way of knowing which target generated a certain measurement. Instead, among any measurements $z_{k}$ that are closer than a threshold $T_{z}$ to some previously found target $\hat{x}\;\in \;\hat{X}$, i.e., $∥g_{k}^{-1}\left(z_{k}\right)-\hat{x}∥\leq T_{z}$, the closest measurement to $\hat{x}$ is removed from $Z_{k}$ (note the transformation of *z* to Cartesian coordinates). Only a single measurement is removed because a target generates at most one measurement.

### 4.3. Obstacle Avoidance

Recall that a low-level controller is used to drive the drone to the next waypoint determined by the planner. Along the trajectory, obstacles may appear. We use a simple and computationally efficient method to avoid detected obstacles. A collision danger indicator between the drone position *q* and the obstacle *o* closest to it is computed as [50]:(14)$$O=\left\{\begin{array}{cc} 1, & ∥q-o∥<d_{l}\\ 0 & \;\mathrm{otherwise}\; \end{array}\right.$$
where $d_{l}$ is the minimum admissible distance to the obstacle.

The control input is then modified by a collision avoidance term, which gets activated whenever O becomes 1:(15)$${\tilde{u}}_{t}=u_{t}-k_{obs}O$$
where $k_{obs}$ is a positive gain. Recall that for the low-level control, we use the time index *t*.

Moreover, if at any point the next waypoint is generated too close to an obstacle, this waypoint is moved to the minimum admissible distance from the obstacle along the line joining the drone and the waypoint.

Algorithm 1 summarizes the target search procedure and integrates the components from Section 4.1, Section 4.2, Section 4.3. Exploration bonus $\iota_{0}$ is 1 everywhere, and the set $\hat{X}$ of found targets is initially empty.
**Algorithm 1** Target search at step *k*  1:get measurements $Z_{k}$ from sensor  2:**for**$\;\hat{x}\;\in \;\hat{X}\;$**do**  3:      $Z_{\mathrm{aux}}=\{z_{k}\in Z_{k}\;|\;∥g^{-1}\left(z_{k}\right)-\hat{x}∥\leq T_{z}\}$ measurements from found target.  4:      **if** $Z_{\mathrm{aux}}$ is nonempty **then**  5:            remove measurement:  6:            $Z_{k}=Z_{k}\setminus {\mathrm{argmin}}_{z_{k}\in Z_{\mathrm{aux}}}∥g_{k}^{-1}\left(z_{k}\right)-\hat{x}∥$  7:      **end if**  8:**end for**  9:run filter from Section 3 with measurements $Z_{k}$10:use K-means to find clusters $C_{i,k},i=1,\ldots ,c_{k}$11:**for** each cluster $i=1,\ldots ,c_{k}$ **do**12:      **if** $r_{i}\leq T_{r}$ and $\sum_{i\;\in C_{k}}\omega_{i}\geq T_{m}$ **then** target found:13:             delete all particles $i\in C_{i,k}$14:             $\hat{X}=\hat{X}\cup \hat{x}$15:      **end if**16:**end for**17:update exploration component $\iota_{k}$ using (Equation 11)18:**for** each $q_{k}$ generated with (Equation 9) **do**19:      compute exploration $E(q_{k})$ (Equation 12) and target refinement $T(q_{k})$ (Equation 13)20:**end for**21:find best sequence $q_{k}^{*}$ with (Equation 10)22:go to the first position $q_{k+1}^{*}=:q_{d}$ while avoiding obstacles with (Equation 15)

## 5. Simulation Results

To validate the efficiency of the proposed target search algorithm, we run seven simulated experiment sets in 3D target space, referred to as E1 through E7. In E1, we investigate the performance of the receding-horizon planner as a function of the horizon τ.

In E2 (for uniformly, sparsely distributed targets) and E3 (for clustered targets), we compare the new algorithm against three baselines. The first baseline is a standard lawnmower that covers the environment uniformly. The other two baselines are representative of the literature on active target searching with intensity-function filtering [7,19,30,32,33,35,36,37,42]: a planner that uses only MI without exploration, like in [33], and one with an unknown-target search component, based on [35]. In [35], the intensity function is represented as a Gaussian mixture, target refinement is achieved by driving to the closest Gaussian, and (in separate experiments) target searching is achieved by driving to the widest Gaussian. A future research direction suggested by the authors of [35] is to mix the two strategies. Here, we adapt their method firstly by applying it to the clusters in our particle-based representation instead of Gaussian components since they have the same meaning of being potential targets. Secondly, we choose a simple mix between search and refinement: the drone first drives to the nearest cluster, then to the widest, then to the nearest, and so on.

Starting with E4, we return to our method that includes exploration. In E4, we evaluate the influence of the cluster width threshold $T_{r}$ on the error between the real and estimated target positions. In E5, we compare MI with center-probability target refinement. Note that all other experiments employ MI in order to align with the existing target search literature. In E6, an experiment is run without any targets to demonstrate how the drone fills the space with an exploratory trajectory similar to that of a lawnmower. Finally, E7 illustrates our obstacle avoidance scheme.

In all these experiments, the 3D environment is E=[0,250]×[0,250]×[0,250] m^3^. The distance from the current position to the next one is set to 12 m, and the candidates δq are six different position choices at this distance, as shown in Figure 3. When a 3D lawnmower is used, the altitude difference between each 2D “lawnmower layer” is constant and is set to 48 m, and the X−Y lawnmower spacing in each such layer is also set to 48 m. This spacing is chosen to be large enough so that the lawnmower is relatively fast but does not run the risk of missing targets.

A Parrot Mambo minidrone model is used [51], which matches the type of drone used in our experiments below, and a linear quadratic controller is designed to move between waypoints. Note that the model has 12 states in reality, but for simplicity at the planning level, we keep the convention that the state is equal to the drone position. The initial position of the drone is $q_{0}={\left[0,0,0\right]}^{T}$.

The parameters of the probability of detection π(x,q) from Section 2 are: G=0.98, $F_{X}=F_{Y}=F_{Z}=25$. The covariance matrix for the sensor noise is R=diag[1.9,0.25,0.41]. We set the maximum number of particles as 5000 to reduce the computation time. The threshold values in Algorithm 1 are set as $T_{r}$ = 1.1 m, $T_{m}$ = 2.2, and $T_{z}$ = 5 m. The value of the parameter α, which trades off exploration with refinement in (Equation 10), is set to 5.

In experiments E1–E3 and E5, we report the mean number of targets detected, along with 95% confidence intervals on the mean out of 10 independent runs. Target positions are randomized in each run. The trajectory length is chosen differently for each experiment so that all algorithms have a chance to detect all the targets.

*E1: Influence of the planner horizon*: We consider 12 targets uniformly distributed at random locations. The trajectory length is chosen to be 330 steps. Figure 4 shows the number of target detections over time for a varying horizon τ=1,2,3. It can be seen that horizon 1 is statistically indistinguishable from 2 and 3. Therefore, we choose horizon τ=1 for the remaining simulations as it is computationally more efficient.*E2: Planner performance for uniformly distributed targets*: We consider 12 targets uniformly and randomly distributed throughout *E*. The trajectory length is chosen to be 812 steps since the lawnmower needs this length to complete the search of the whole space. The length is the same for all algorithms for fairness.

**Figure 4 sensors-24-02868-f004:**
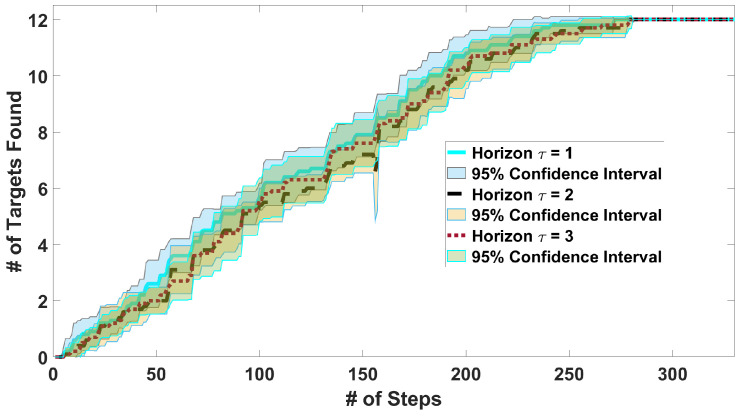
Average number of targets detected for targets placed uniformly at random locations on 10 maps with 95% confidence intervals on the mean using horizons τ=[1,2,3].

The results in Figure 5 show the number of target detections as a function of the number of measurement samples. Our proposed planner works better than the three baselines: lawnmower, the MI-only method adapted from [33], and the method adapted from [35]. The lawnmower is much slower than our planner, whereas the performance of the method adapted from [35] is between that of the lawnmower and that of our planner. Finally, MI-only does not find all the targets but focuses on those targets that happen to be detected. Note that instead of choosing constantly spaced waypoints, the method adapted from [35] drives to arbitrarily far away clusters. To make the comparison fair, along the trajectories, this method measures at intervals equal to the waypoint spacing in our method, and the horizontal axis represents, in this case, the number of measurements.

Figure 6 shows all the algorithms’ trajectories in one of the 10 experiments. Thanks to the inclusion of the exploration component, our algorithm finds all the targets, unlike the MI-only method. The lawnmower covers the 3D environment uniformly, so it finds all the targets but more slowly than our method since we focus on relevant regions first. The method adapted from [35] also focuses on relevant regions, so it is faster than the lawnmower, but as already shown in Figure 5, it is still slower than our method.

Regarding computation time, our new method takes 0.041s to plan, the MI-only approach takes 0.012 s, the lawnmower takes 0.005 s, and the method adapted from [35] takes 0.011s, where all the times are averages across waypoints. The computer used is an HP (HP Inc., Palo Alto, CA, USA) equipped with an Intel 1165G7 CPU and 16 GB of RAM, running MATLAB R2023. The relationship between computation times is to be expected due to the higher complexity of the exploration-based planner compared to the baselines. Recall also that our main goal is to reduce the length of the trajectory of the robot in the environment since, in practice, real-world robot motion and sensor samples are often much more costly than computation.

*E3: Planner performance for clustered targets*: We consider 12 targets placed in 2 clusters of 6 targets, each at a random location. The trajectory length is the same as for E2. Figure 7 (top) shows the number of targets detected over time. We see that the performance of our algorithm is again better than those of the three baselines. Figure 7 (bottom) shows the positions of the actual targets as well as the target locations estimated by our method. The root mean squared error (RMSE) between the actual and estimated positions is 3.14 m, which is relatively small for a domain with a size of $250^{3}$ m^3^. This RMSE value depends on the covariance of the Gaussian noise in the sensor model (Equation 4) and the threshold values in Algorithm 1. For instance, we can reduce the error by making the cluster width threshold $T_{r}$ smaller, as we show in the next experiment.*E4: Threshold value versus RMSE*. For 12 targets uniformly distributed at random locations, we used 20 different values of the cluster radius threshold $T_{r}$ that varied in range from 0.5 to 2.4. The results in Figure 8 (left) show the RMSE values between the actual and estimated target locations as a function of the threshold value $T_{r}$. Errors are directly related to threshold values and can be made smaller by reducing $T_{r}$. Doing this, of course, increases the number of steps taken by the drone to find all the targets, as shown in Figure 8 (right).*E5: Target refinement with center probabilities*: In this experiment, we compare the performance when the target refinement component is MI versus when the center-probabilities version (Equation 13) is used. Exploration is included. We consider 12 uniformly and randomly distributed targets. The trajectory length is 330 steps. Figure 9 shows the number of targets detected over time. The algorithm found all targets in a nearly equal amount of steps using the two options for target refinement. The main difference is in computational time: the MI-based algorithm takes an average of 0.041 s per step to plan, while using center probabilities is faster, with an average of 0.018 s per step.
Figure 7**Top**: Average number of clustered targets detected in 10 random maps. **Bottom**: Estimation error using our method.
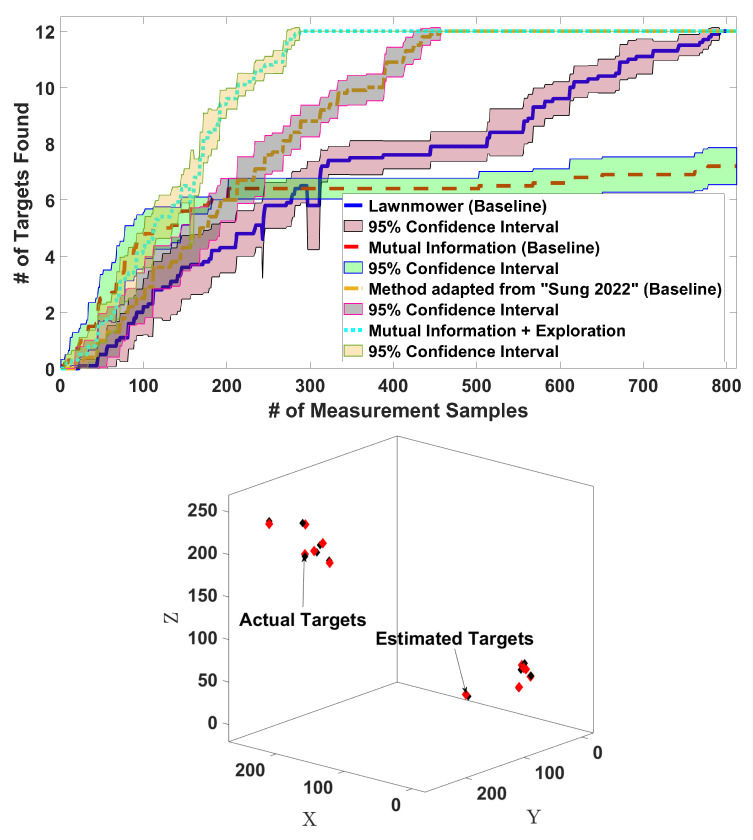

Figure 8Results for E4. **Left:** Target position error for different thresholds. **Right:** Number of steps taken by the drone to find all targets.
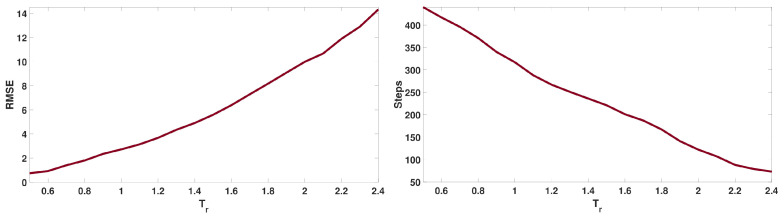

Figure 9Detected average number of targets with MI and the center-probabilities methods in 10 random maps.
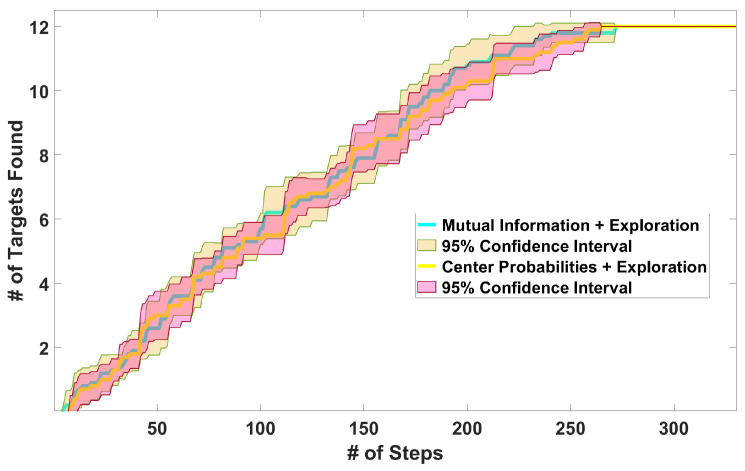

*E6: Trajectory with no targets*: We show in Figure 10 (left) how the drone explores the environment in the absence of targets. The trajectory length is chosen as 600 steps. The drone flies similarly to a lawnmower pattern because in the absence of target measurements, the exploration component drives it to cover the whole environment.*E7: Obstacle avoidance*: We consider 12 targets and 5 obstacles with various sizes placed manually at arbitrary locations, as shown in Figure 10 (right). The trajectory length is 380 steps. For obstacle avoidance, dl is set to 11 m, and kobs=0.09 is used in (Equation 15). Figure 10 (right) shows the drone searching for the targets while avoiding the obstacles. It takes about 380 steps to find all the targets, compared to 330 steps without obstacles.

Next, we present our hardware setup and experimental results.

## 6. Experimental Results

In this section, real-life implementation results are given using a Parrot Mambo drone. Since creating real, free-floating static targets would be difficult, for this experiment, the problem is reduced to a 2D search with targets on the floor and the drone flying at a constant altitude.

We start with an overview of the experimental framework in Figure 11. After getting the measurements, we update the target intensity function, and the planner generates a new reference position for the drone. However, the Parrot Mambo minidrone has little memory and computation power, so it is not possible to do all of the processing onboard. To solve this problem, we created two different nodes: one deployed on the drone, and the second used remotely on the computer. To share information between these nodes, we establish a UDP communication channel. On the minidrone, we implement low-level segmentation of the images and perform a coordinate transformation to get the measurements, which are then transmitted to the host. On the computer, we run the filter and the planner to generate the each next waypoint. This waypoint is transmitted to the minidrone and is tracked using the built-in controller of the minidrone.

Next, we describe the hardware and low-level software in Section 6.1. The high-level setup for the experiment as well as the real-world results illustrating the performance of our algorithm are described in Section 6.2.

### 6.1. Hardware, Sensing, and Control

The Parrot Mambo is a quadcopter that can be commanded through MATLAB 2023 via Bluetooth. Figure 12 illustrates the drone searching for unknown targets. The drone is equipped with an inertial measurement unit (IMU) containing a three-axis accelerometer and a three-axis gyroscope, ultrasound and pressure sensors for altitude measurements, and a downward-facing camera that has a 120 × 160 pixel resolution at 60 frames per second, which is useful for optical flow estimation and image processing [52]. The MATLAB/Simulink support package version 12 [8] allows access to the internal sensor data and deployment of control algorithms and sensor fusion methods in real-time.

We placed several blue markers that represent targets on the ground. Through the use of the drone-mounted camera, pictures containing these markers are taken and fed into an image processing algorithm for the identification and localization of the markers. The FOV of the camera depends on the drone’s altitude, and to keep the FOV the same size as well as to ensure that the targets are reasonably sized, the drone flies at a constant altitude of 1 m above the ground.

Given an RGB image (R,G,B) for which the three channels R,G,B are each an n×m dimensional matrix, image segmentation is used to identify the blue markers. A simple way of doing this is through thresholding [53]. First, to enhance the blue color, we compute matrix $O_{p}=B-\frac{R}{2}-\frac{G}{2}$, where operations are applied element-wise. After applying a single-level threshold of value $T_{im}$, the resulting binary image $T_{im}\in {\left\{0,1\right\}}^{n\times m}$ is:(16)$$\begin{array}{c} T_{im}\left(x_{im},y_{im}\right)=\left\{\begin{array}{cc} 0, & O_{p}\left(x_{im},y_{im}\right)<T_{im}\\ 1, & O_{p}\left(x_{im},y_{im}\right)>T_{im} \end{array}\right. \end{array}$$
An example of thresholding is shown in Figure 13, with the threshold $T_{im}=35$.

After segmentation, we compute the homogeneous coordinates $P_{im}={\left[\begin{array}{ccc} x_{im} & y_{im} & 1 \end{array}\right]}^{T}$ of the segmented target’s centroid in the image plane. A backward projection is performed to find the world-frame homogeneous coordinates $P_{w}={\left[\begin{array}{cccc} x_{w} & y_{w} & z_{w} & 1 \end{array}\right]}^{T}$ of the target. Here, the targets are on the floor, so $z_{w}$ is taken as zero [52]. Then, $P_{w}$ is computed by solving:(17)$$\begin{array}{c} P_{im}=C\Lambda P_{w} \end{array}$$
where Λ defines the transformation between the world and camera frames and is defined using the position of the drone $\left(x_{q},y_{q},z_{q}\right)$ centered on the FOV, and C is the camera’s intrinsic matrix, which has the standard form:(18)$$C=\left[\begin{array}{cccc} \beta_{x} & 0\; & x_{ic} & 0\\ 0 & \beta_{y}\; & y_{ic} & 0\\ 0 & 0 & 1 & 0 \end{array}\right]$$
where $\beta_{x}$ and $\beta_{y}$ represent the pixel focal length, and $\left(x_{ic},y_{ic}\right)$ are the optical center coordinates expressed in pixels. By finally transforming $x_{w},y_{w}$ into a bearing and range, we obtain the target measurement. The entire pipeline constructed above to output this measurement constitutes our sensor.

For low-level control, we use the built-in altitude and position controllers from the Simulink model provided by MathWorks (Natick, MA, USA) [8].

### 6.2. High-Level Setup and Results

The targets (blue markers) are located in a 2D environment belonging to the floor plane and having the shape [−0.3,2] m×[−0.3,2] m. The planner considers eight possible position changes: forward, backward, right, left, and the diagonals, each shifting the position by 0.2 m.

Due to the particularities of the hardware experiment, the sensor model is different from the one in Section 2: see again the perception pipeline in Section 6.1. In particular, the detection probability is:(19)$$\pi \left(x,q\right)=\left\{\begin{array}{cc} 1 & x_{ik}\;\in \;F\\ 0 & x_{ik}\;\notin \;F \end{array}\right.$$
where $F=\left[X_{q}-0.2,X_{q}+0.2\right]\;\;\times \;\left[Y_{q}-0.2,Y_{q}+0.2\right]$ is the camera’s FOV at the 1 m altitude of the drone. Target measurements have a form similar to (Equation 4):(20)$$Z_{k}=\underset{i\;\in \left\{1,\ldots ,N_{k}\right\}\;s.t.\;b_{ik}=1}{⋃}\left[{\hat{g}}_{ik}\left(x_{ik}\right)+{\hat{\varrho }}_{ik}\right]$$
but now, ${\hat{g}}_{ik}={\left[d_{ik},\theta_{ik}\right]}^{T}$ contains only the Euclidean distance $d_{ik}$ between the 2D positions $q_{k}$ and $x_{ik}$ and the bearing angle $\theta_{ik}$, which is computed as in (Equation 4). Moreover, ${\hat{\varrho }}_{ik}$ is 2D Gaussian noise. To estimate its covariance *R*, we captured 100 images of a single target from different positions of the drone and then computed the empirical covariance of these measurements, obtaining R=diag[0.016,0.052]. Note that due to the binary π, $b_{ik}=1$ if and only if $x_{ik}\in F$. We set α=5.

The filter and planner equations from Section 3 and Section 4 hold in 2D, with the mention that the exploration bonus ι is now interpolated bilinearly on a 2D grid.

To validate our algorithm, we initialize the drone at coordinates ${\left[0,0\right]}^{T}$. We set the maximum number of particles to 5000 to reduce the computation time. The threshold values in Algorithm 1 are set experimentally to $T_{r}=0.1$,$T_{m}=0.2$, and $T_{z}=0.3$. Up to 100 s is allocated to complete the experiment. The high-level sampling period used for estimation, generating new waypoints, and communication is 3.05 s.

We consider 12 targets manually placed at arbitrary locations. Figure 14 (left) shows our algorithm’s trajectory, together with the actual and estimated target locations. The drone finds the targets with an RMSE between the actual and estimated target locations of 0.08 m, which is relatively small. There is some drift because we rely on the onboard sensors. A video of a real-world experiment is available online at http://rocon.utcluj.ro/files/mambo_targetsearch.mp4 (accessed on 27 July 2023).

Figure 14 (right) shows the number of targets detected using our method compared to real-life implementations of the lawnmower and MI-only baselines. The X-Y lawnmower spacing in each such layer is set to 0.2 m. Like in the simulations, the proposed method works better than the lawnmower and MI-only methods.

## 7. Conclusions

In this paper, we addressed the problem of finding an unknown number of static targets using a drone. The proposed method is composed of a multi-target filter, a planner based on exploration and target refinement, and a method to remove found targets. The effectiveness of the proposed algorithm was validated through extensive simulations and in a real-robot experiment. The results demonstrate that the proposed algorithm is better than the two active-search baselines and the lawnmower approach.

Our solution has several limitations that should be addressed in future work. Control costs, such as the energy consumed by the drone, are not taken into account, and we aim to include such costs in the planning objective. Importantly, our experiments were conducted indoors, where weather conditions were not a factor. Outdoor experiments introduce nontrivial challenges such as weather-resilient target detection in images, for which the methods from [54,55] could be used. Finally, our algorithm assumes that the drone’s pose is known; joint filtering of target and pose measurements can be performed, as in [56].

## Figures and Tables

**Figure 1 sensors-24-02868-f001:**
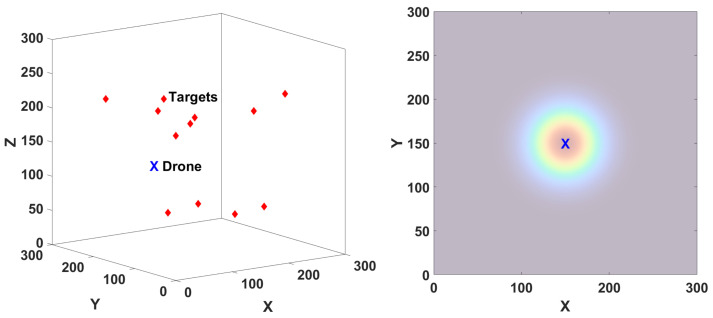
**Left**: 3D space with 12 targets and a drone. **Right**: drone with a spherical field of view that is symmetrical in all three axes. The dark-orange-to-blue colors show the probability of observation (higher-to-lower) at the current position of the drone. This is a 2D slice of the 3D probability of observation.

**Figure 2 sensors-24-02868-f002:**
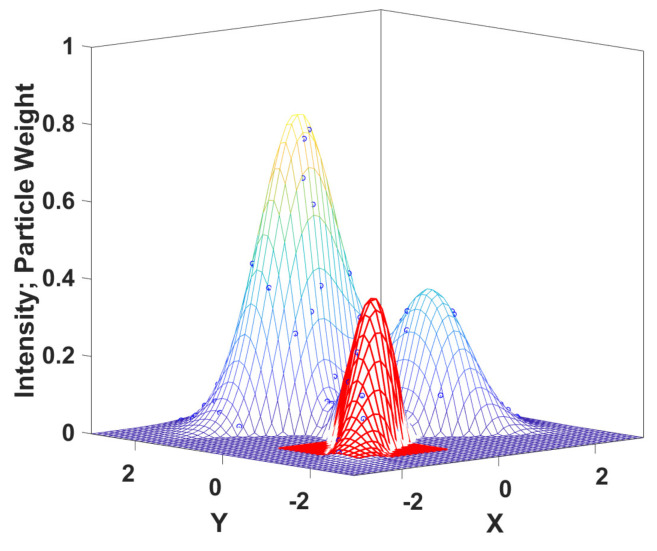
Illustration of an intensity function defined over 2D space.

**Figure 3 sensors-24-02868-f003:**
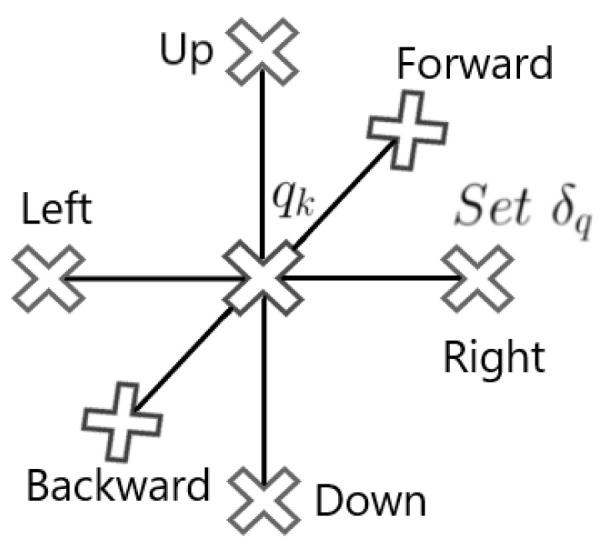
Position changes δq for the simulations in Section 5.

**Figure 5 sensors-24-02868-f005:**
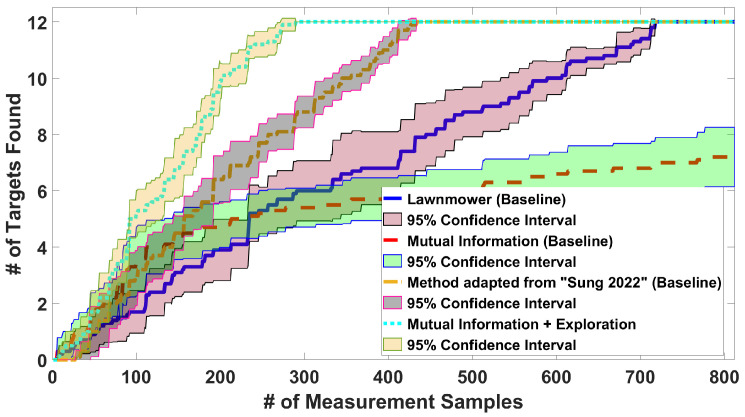
Average number of targets detected on 10 maps with our algorithm, the lawnmower baseline, the MI-only baseline adapted from [33], and the baseline adapted from [35].

**Figure 6 sensors-24-02868-f006:**
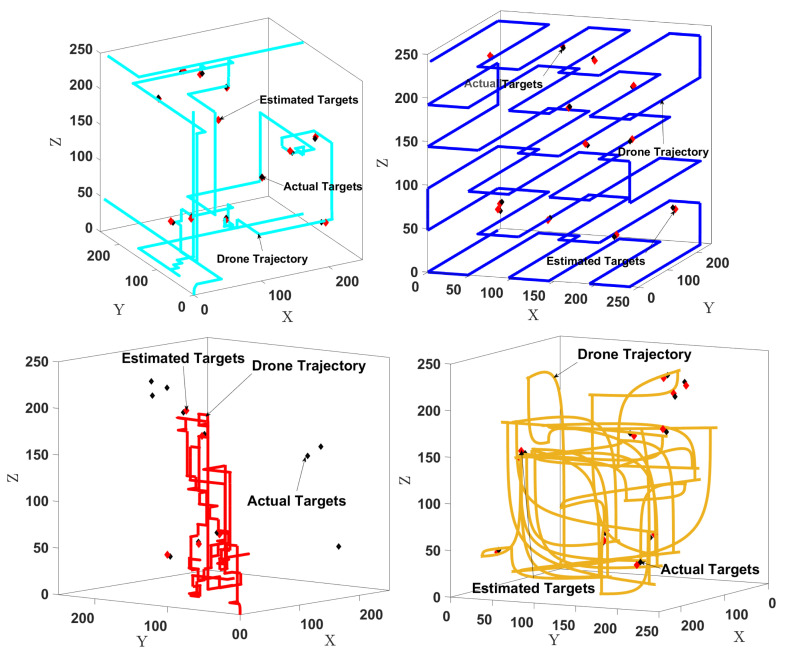
Examples of trajectories with the actual targets (black diamonds) and estimated targets (red diamonds). **Top left**: Our method (mutual information with exploration). **Top right**: Lawnmower baseline. **Bottom left**: MI-only baseline adapted from [33]. **Bottom right**: Baseline adapted from [35].

**Figure 10 sensors-24-02868-f010:**
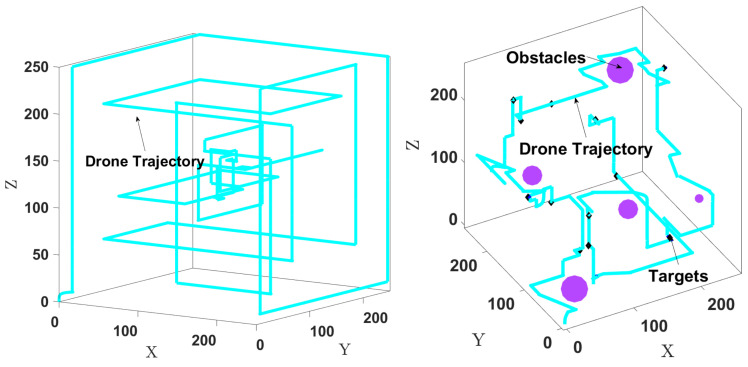
**Left**: Drone trajectory when there is no target. **Right**: Drone avoiding obstacles while searching for targets.

**Figure 11 sensors-24-02868-f011:**
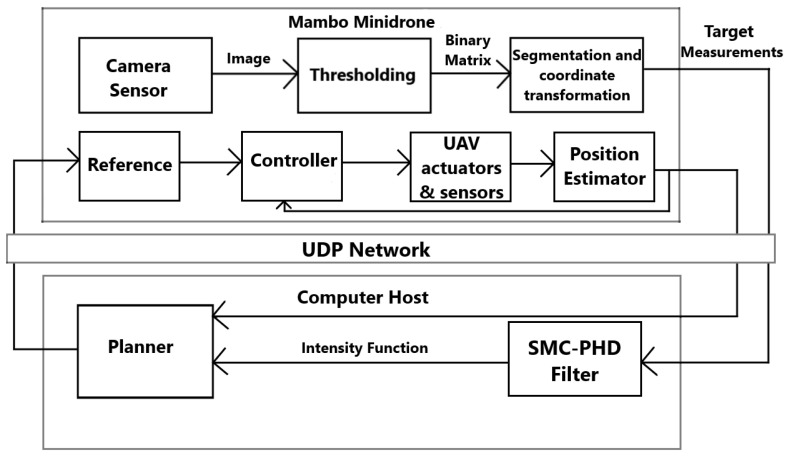
Block diagram of the Parrot Mambo application.

**Figure 12 sensors-24-02868-f012:**
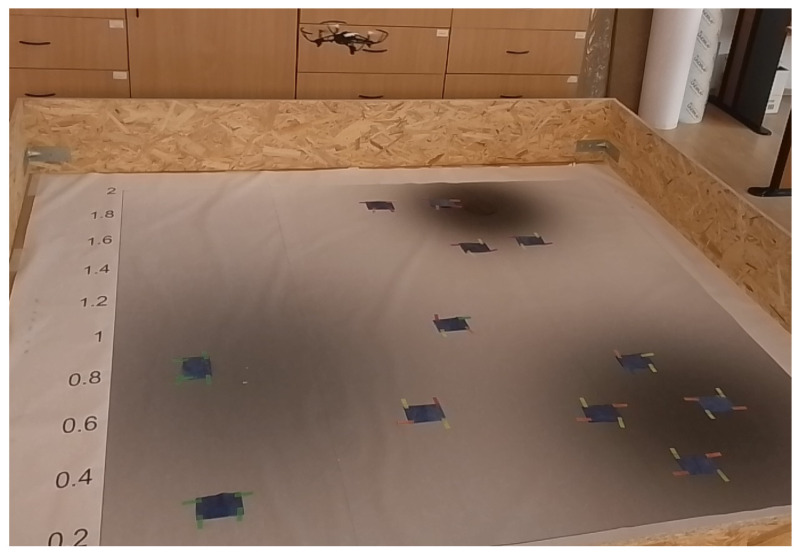
Parrot Mambo minidrone searching for targets (blue markers).

**Figure 13 sensors-24-02868-f013:**
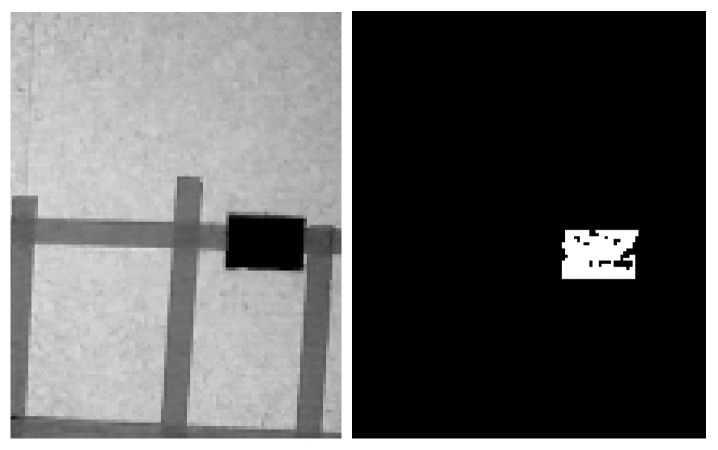
Example of blue color thresholding. **Left**: before thresholding. **Right**: after thresholding.

**Figure 14 sensors-24-02868-f014:**
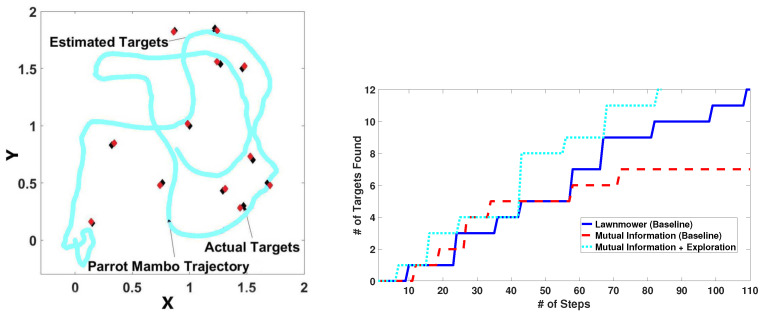
**Left**: Trajectories with our algorithm in 2D real-world with actual and estimated targets. **Right**: Number of targets detected using real Parrot Mambo minidrone compared to the lawnmower and MI-only baselines.

## Data Availability

Data are contained within the article.

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
