# Peer review of "Exploration-Based Planning for Multiple-Target Search with Real-Drone Results"

_sensors, 2024, doi:10.3390/s24092868_

Round 1

Reviewer 1 Report

Comments and Suggestions for Authors

This paper has proposed a novel receding-horizon planner that selects the next position of a drone by maximizing an objective that combines exploration and target refinement. More specifically, confidently localized targets are saved and removed from consideration along with their future measurements. A state feedback controller with an obstacle-avoidance component is used to reach the desired waypoints. We demonstrate the performance of our approach through a series of simulations, as well as in a real-robot experiment with a Parrot Mambo drone, for the case when targets are located on the floor.

The detailed comments are summarized as follows:

1. The abstract section should be enriched. The detailed experimental results should be incorporated in the abstract section.

2. The main contributions of this work should be summarized as several points in a bullet-point fashion.

3. This paper focuses on exploration-based search. How is the performance of the proposed method under inclement weather conditions? How to deal with these situations? I suggest that the authors at least consider these potential situations and give some feasible solutions. Some recent image enhancement and restoration methods (1-4) can be used as the pre-processing step to handle them to improve the quality of the captured images.

1) Fast haze removal for nighttime image using maximum reflectance prior

2) Multi-purpose Oriented Single Nighttime Image Haze Removal Based on Unified Variational Retinex Model

3) Nighttime dehazing with a synthetic benchmark

4) Nighthazeformer: Single nighttime haze removal using prior query transformer

4. The limitations of the proposed search method should be discussed in depth.

5. From the experimental results, I understand that the proposed method can achieve good performance. However, how to demonstrate the superiority of the proposed method when compared to existing methods? Could you provide more recent state-of-the-art methods for comparisons?

Based on the above comments, I recommend major revisions for this manuscript.

Comments on the Quality of English Language

Moderate editing of English language required

Author Response

Thanks for your valuable time to review the manuscript, it helps us alot to improve the quality of the paper.

Please find the attached file of the response of your review.

Reviewer 2 Report

Comments and Suggestions for Authors

Reviewer’s comments to the manuscript “Exploration-Based Planning for Multiple-Target Search with

Real-Drone Results" (Authors: Bilal Yousuf, Zsófia Lendek and Lucian Busoniu).

 The article is devoted to a novel receding-horizon planner that selects the next position of a drone by maximizing an objective that combines exploration and target refinement. Confidently localized targets are saved and removed from consideration along with their future measurements. A state feedback controller with an obstacle-avoidance component is used to reach the desired waypoints. The work demonstrates the performance of approach through a series of simulations, as well as in a real-robot experiment with a Parrot Mambo drone, for the case when targets are located on the floor. The novel framework works better than lawnmowers and then using only target refinement in the planning objective.

 There are some other points to correct or to make the information more exact:

 Essential drawbacks.

Remark 1. The authors of the article have already published work on this topic [https://doi.org/10.1016/j.ifacol.2022.07.614]. The work has already presented the described path planning algorithm as a novel method. What is the significant difference between the already published materials and those presented in this article?

Remark 2. Currently, the deep learning methods are actively used in solving the problems described by the authors, for example in works [https://doi.org/10.3390/drones7090572, https://doi.org/10.3389/fnbot.2023.1302898]. What advantage does the algorithm presented by authors have over deep learning-based methods? The article does not contain a comparison with these algorithms.

 Technical drawbacks.

Remark 1. Lines 261-264. The text goes beyond the boundaries.

Comments on the Quality of English Language

The article contains punctuation errors, in particular the uncorrected use of commas.

Author Response

Thanks for your positive opinions and suggestion to improve the quality of the paper, please find the attach Pdf of our response to your review

Reviewer 3 Report

Comments and Suggestions for Authors

The paper states that the goal is to find as quickly as possible an unknown number of static targets at unknown positions, which in itself is meaningless. In general, the problem of searching in a given area in the absence of information about the location of the targets is formulated as a task of maximizing the search potential. The definition of which includes current measurements, information about search strategies, information about interference, errors and uncertainties of the model.

The optimization problem (10) includes an unknown alpha parameter and unknown detection probabilities that cannot be selected from the essence of the applied problem, since in this case the experiments are not repeatable.

As a result, the algorithm proposed by the authors allows us to solve the problem of planning a route, avoiding obstacles while simultaneously searching for objects. However, the scientific novelty of the work is in great doubt. It is necessary to clarify the formulation of the applied problem, the mathematical formulation of the main problem, the novelty of individual subtasks of searching and avoiding obstacles.

Please see links to the articles

A Real-Time Path-Planning Algorithm based on Receding Horizon Techniques M. Murillo · G. S´anchez · L. Genzelis · L. Giovanini https://doi.org/10.1007/s10846-017-0740-1

Explore Locally, Plan Globally: A Path Planning Framework for Autonomous Robotic Exploration in Subterranean Environments Tung Dang, Shehryar Khattak, Frank Mascarich, Kostas Alexis DOI: 10.1109/ICAR46387.2019.8981594

A comparison of path planning strategies for autonomous exploration and mapping of unknown environments Miguel Juliá · Arturo Gil · Oscar Reinoso DOI 10.1007/s10514-012-9298-8

Receding horizon path planning for 3D exploration and surface inspection Andreas Bircher · Mina Kamel · Kostas Alexis · Helen Oleynikova·Roland Siegwart  https://doi.org/10.1007/s10514-016-9610-0

Author Response

Thanks for the quick and very critical review. it helps us to improve the paper quality. Please find the attach file of our response to your review.

Round 2

Reviewer 1 Report

Comments and Suggestions for Authors

I recommend accept for publication.

Comments on the Quality of English Language

Moderate editing of English language required

Reviewer 2 Report

Comments and Suggestions for Authors

The authors took into account the comments; there are no new comments.